# Hierarchical Multi-Agent Skill Discovery

**Mingyu Yang**[1], **Yaodong Yang**[2][†], **Zhenbo Lu**[3][†], **Wengang Zhou**[1,3], **Houqiang Li**[1,3]

[1]University of Science and Technology of China, [2]Institute for AI, Peking University
[3]Institute of Artificial Intelligence, Hefei Comprehensive National Science Center
ymy@mail.ustc.edu.cn, yaodong.yang@pku.edu.cn
luzhenbo@iai.ustc.edu.cn, {zhwg,lihq}@ustc.edu.cn

## Abstract

Skill discovery has shown significant progress in unsupervised reinforcement learning. This approach enables the discovery of a wide range of skills without any extrinsic reward, which can be effectively combined to tackle complex tasks. However, such unsupervised skill learning has not been well applied to multi-agent reinforcement learning (MARL) due to two primary challenges. One is how to learn skills not only for the individual agents but also for the entire team, and the other is how to coordinate the skills of different agents to accomplish multi-agent tasks. To address these challenges, we present Hierarchical Multi-Agent Skill Discovery (HMASD), a two-level hierarchical algorithm for discovering both team and individual skills in MARL. The high-level policy employs a transformer structure to realize sequential skill assignment, while the low-level policy learns to discover valuable team and individual skills. We evaluate HMASD on sparse reward multi-agent benchmarks, and the results show that HMASD achieves significant performance improvements compared to strong MARL baselines.

## 1   Introduction

Multi-agent reinforcement learning (MARL) has recently demonstrated remarkable potential in solving various real-world problems, such as unmanned aerial vehicles [1], autonomous driving [2] and traffic light control [3]. Despite its broad applications, current MARL algorithms [4, 5] typically require well-crafted team or individual rewards to guide the agents to learn policies for efficient coordination. This limitation hinders the generalization of MARL to the sparse reward multi-agent tasks, where agents receive a non-zero reward only when they coordinate to achieve a challenging goal. Compared to single-agent tasks, the sparse reward problem in multi-agent tasks poses more challenges. On the one hand, the joint state and action spaces of multi-agent tasks increase exponentially with the number of agents, which exacerbates the difficulty for agents to explore those valuable but rare states. On the other hand, we need to distribute the received sparse reward not only to different timesteps but also to different agents [6].

To solve sparse reward multi-agent problems, a promising approach is to discover underlying skills within the multi-agent task and effectively combine these skills to achieve the final goal. For example, in a football game, there are various skills involved such as dribbling, passing, receiving and shooting. A common training strategy for a football team is to let each player learn these fundamental skills first and then train players together to cooperatively use these skills for scoring the goal, which is more efficient than directly training players without any skills together. Similarly, in a sparse reward multi-agent task, it's difficult for agents to cooperatively achieve a challenging goal from scratch. We can first try to discover the underlying useful skills within the task, which is much easier than solving the entire task under the sparse reward setting [7, 8]. Although these discovered skills may

---

[†]Corresponding authors: Yaodong Yang and Zhenbo Lu

not independently achieve the goal or induce non-zero rewards, we can learn to effectively combine them to accomplish the final task. In short, it is possible to decompose a sparse reward multi-agent task into a combination of different skills, which greatly simplifies the task complexity.

Previous works mainly follow two paradigms to discover skills in MARL. One is to let all agents learn a shared team skill [9, 10], which promotes team cooperation behaviors but suffers from high complexity. The other does the opposite and learns skills only from the perspective of individual agents [11, 12], which is more efficient since individual skills are usually easier to learn than team skills. However, merely learning individual skills may be insufficient to achieve team objectives.

Taking football as an example again, individual skills can refer to the technical abilities of individual players, such as dribbling and passing, while team skills refer to the ability of players to work together as a whole, *i.e.*, the overall team tactics, such as wing-play and tiki-taka [13]. As shown in Fig. 1, the goal of a football team is to let all players coordinate to master useful team skills from a global perspective. Directly learning the team skill is usually too complex. A better way is to decompose the team skill into different individual skills for players from an individual perspective, and ensure that the joint behavior of all players can form the team tactic. Both team skills

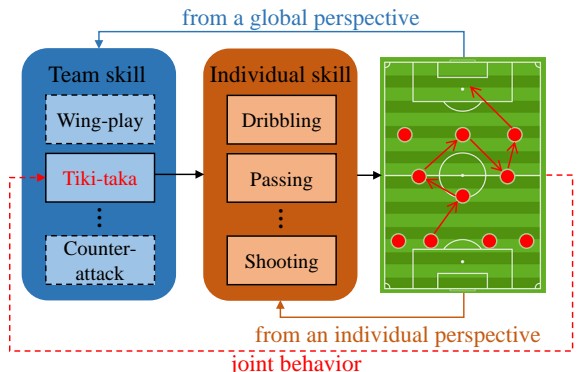

Figure 1: Illustration of team skill and individual skill in football.

and individual skills are important to a successful team. Therefore, a primary challenge for multi-agent skill discovery is how to simultaneously learn the individual skill for each individual agent and the team skill for the whole team, and the second challenge is how to combine these skills to accomplish multi-agent tasks.

To this end, we present a new paradigm for discovering both team and individual skills in MARL. Inspired by probabilistic inference in RL [14], we embed the multi-agent skill discovery problem into a probabilistic graphical model (PGM). With the PGM, we can formulate multi-agent skill discovery as an inference problem, allowing us to take advantage of various approximate inference tools. Then, we derive a variational lower bound as our optimization objective by following the structured variational inference [14], and propose Hierarchical Multi-Agent Skill Discovery (HMASD), a practical two-level hierarchical MARL algorithm for optimizing the derived lower bound. Specifically, the high-level policy employs a transformer [15] structure to assign skills to agents, with inputs consisting of the sequence of global state and all agents' observations, and outputs consisting of the sequence of team skill and agents' individual skills. In this way, an agent's individual skill depends on the team skill and all previous agents' individual skills, allowing agents to choose complementary skills and achieve better skill combination. The low-level policy chooses primitive actions for each agent to interact with the environment conditioned on the assigned skills. Moreover, we introduce two skill discriminators to generate intrinsic rewards for agents to learn diverse and distinguishable skills. Finally, we show that HMASD achieves superior performance on sparse reward multi-agent benchmarks compared to strong MARL baselines. To our best knowledge, our work is the first attempt to model both team skills and individual skills with the probabilistic graphical model in MARL.

## 2 Preliminaries

In this section, we first introduce the problem formulation and notations for cooperative MARL. Then, we describe the concept of skill and a mutual information based objective for skill discovery in unsupervised RL. We finally present the framework of representing RL as a probabilistic graphical model. More related works are discussed in Sec. 5.

### 2.1 Problem Formulation

In this work, we consider a fully cooperative MARL problem, which is usually described as a decentralized partially observable markov decision process (Dec-POMDP) [16]. A Dec-POMDP is

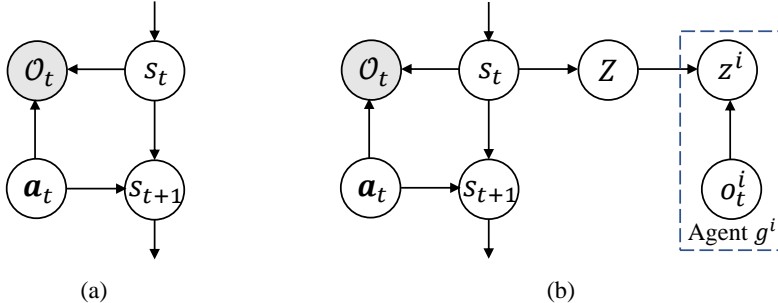

(a)                                                              (b)

Figure 2: (a) RL can be cast as an inference problem by considering a basic probabilistic graphical model consisting of states $s_t$, actions $\boldsymbol{a}_t$ and optimality variables $\mathcal{O}_t$. (b) Similarly, we formulate multi-agent skill discovery as an inference problem by augmenting the basic probabilistic graphical model with team skills $Z$, individual skills $z^i$ and observations $o_t^i$.

defined by a tuple $G = \langle \mathcal{N}, \mathcal{S}, \mathcal{A}, P, r, \Omega, O, \gamma \rangle$, where $\mathcal{N} \equiv \{g^1, g^2, \cdots, g^n\}$ is the set of $n$ agents and $\mathcal{S}$ is the global state space of the environment. All agents share the same action space $\mathcal{A}$ and the joint action space is $\mathcal{A}^n$. At timestep $t$, each agent $g^i \in \mathcal{N}$ chooses an action $a_t^i \in \mathcal{A}$, where $i \in \{1, 2, \cdots, n\}$ is the agent identity. The actions of all agents form a joint action $\boldsymbol{a}_t \in \mathcal{A}^n$. By executing $\boldsymbol{a}_t$, the environment transitions to the next global state $s_{t+1} \sim P(s_{t+1}|s_t, \boldsymbol{a}_t)$, and all agents receive a shared team reward $r(s_t, \boldsymbol{a}_t)$. Each agent $g^i$ can only observe a partial observation $o_t^i \in \Omega$ according to the observation probability function $O(s_t, g^i)$. The joint observation of all agents is denoted as $\boldsymbol{o}_t$. $\gamma \in [0, 1)$ is the discount factor. The objective is to learn a joint policy $\boldsymbol{\pi}$ that maximizes the expected global return $\mathbb{E}\left[\sum_{t=0}^{\infty} \gamma^t r_t \,|\, \boldsymbol{\pi}\right]$. In particular, this paper focuses on the sparse reward setting, where the team reward $r_t$ is zero for most timesteps.

## 2.2 Mutual Information based Skill Discovery

Skill discovery is a popular paradigm in unsupervised RL, which enables agents to discover diverse skills without a reward function. A *skill* is represented by a latent variable $z$, which is used as an additional input to the policy and results in a latent-conditioned policy $\pi(a|s, z)$. The intention behind skill discovery is to expect the skill to control which states the agent visits and different skills to visit different states. This can be achieved by a simple objective of maximizing the mutual information between state $s$ and skill $z$, *i.e.*, $\mathcal{I}(s; z) = \mathcal{H}(z) - \mathcal{H}(z|s)$. It's challenging to directly optimize the mutual information. Therefore, a majority of skill discovery methods [17–20] derive a variational lower bound for the mutual information as follows:

$$\mathcal{I}(s; z) = \mathbb{E}_{s,z \sim p(s,z)}\left[\log p(z|s) - \log p(z)\right] \geq \mathbb{E}_{s,z \sim p(s,z)}\left[\log q_\phi(z|s) - \log p(z)\right], \quad (1)$$

where $q_\phi(z|s)$ is a learned model parameterized by $\phi$ to approximate the intractable posterior $p(z|s)$. Then, we can treat $\log q_\phi(z|s) - \log p(z)$ as the intrinsic reward to optimize the latent-conditioned policy $\pi(a|s, z)$, where different $z$ correspond to different skills.

## 2.3 Probabilistic Graphical Model for RL

The probabilistic graphical model (PGM) is a powerful tool for modeling complex and uncertain systems. It provides a graphical representation of the relationship between variables in a system, where nodes correspond to random variables and edges represent conditional dependencies. In recent years, PGM has been widely used in RL to model complex decision-making tasks [14, 21–24]. In this work, we follow the basic PGM for RL [14], which embeds the control problem into a graphical model and formulates it as an inference problem. As shown in Fig. 2(a), it first models the relationship among states, actions, and next states based on the dynamics $P(s_{t+1}|s_t, \boldsymbol{a}_t)$. To incorporate the reward function, it introduces a binary random variable $\mathcal{O}_t$ called *optimality variable* into the model, where $\mathcal{O}_t = 1$ denotes timestep $t$ is optimal, and $\mathcal{O}_t = 0$ indicates timestep $t$ is not optimal. The probability distribution over $\mathcal{O}_t$ is $p(\mathcal{O}_t = 1|s_t, \boldsymbol{a}_t) = \exp\left(r(s_t, \boldsymbol{a}_t)\right)$. Refer to [14], we then perform structured variational inference to derive the final objective, which is to optimize a variational lower bound (also called evidence lower bound). The evidence is that $\mathcal{O}_t = 1$ for all $t \in \{0, \cdots, T\}$. We will use $\mathcal{O}_t$ to denote $\mathcal{O}_t = 1$ for conciseness in the remainder of this paper.

The variational lower bound is given by:

$$\log p(\mathcal{O}_{0:T}) \geq \mathbb{E}_{s_{0:T}, \boldsymbol{a}_{0:T} \sim q(s_{0:T}, \boldsymbol{a}_{0:T})} \left[ \sum_{t=0}^{T} r(s_t, \boldsymbol{a}_t) - \log q(\boldsymbol{a}_t | s_t) \right], \tag{2}$$

where $q(\boldsymbol{a}_t | s_t)$ is the learned policy. Optimizing this lower bound corresponds to maximizing the cumulative reward and the policy entropy at the visited states, which differs from the standard RL objective that only maximizes reward. The entropy term can promote exploration and prevent the policy from becoming too deterministic. This type of RL objective is sometimes known as maximum entropy RL [25, 26].

## 3 Method

In this section, we present our solution for learning both team and individual skills in MARL. We first model the multi-agent skill discovery problem with a PGM and derive a tractable variational lower bound as the optimization objective. We then propose a practical MARL algorithm to optimize the derived lower bound.

### 3.1 Multi-Agent Skill Discovery as an Inference Problem

In this work, we study the skill discovery problem in multi-agent tasks. One way for multi-agent skill discovery is to treat all agents as one big virtual agent and directly learn skills from the perspective of the whole team, which can improve teamwork but suffer from high complexity. Another way is to learn skills from the perspective of each individual agent, which reduces the complexity but lacks collaboration. To combine the advantages of two ways, we propose to learn skills not only from the individual perspective but also from the team viewpoint.

Specifically, we utilize a latent variable denoted by $Z \in \mathcal{Z}$ to represent the skill of the entire team, referred to as team skill. And we use a latent variable denoted by $z^i \in \mathcal{X}$ to represent the skill of agent $g^i$, referred to as individual skill. The individual skills of all agents are represented by $z^{1:n} \in \mathcal{X}^n$. In this work, both the team skill space $\mathcal{Z}$ and the individual skill space $\mathcal{X}$ are defined as discrete spaces consisting of finite latent variables with one-hot encoding, where the number of team skills and individual skills are denoted as $n_Z$ and $n_z$, respectively. The team skill $Z$ is acquired from the global view and is expected to control the global states that the whole team visits, while the individual skill $z^i$ is developed through an individual perspective and is intended to control the partial observations accessed by agent $g^i$. Besides, since individual behaviors are typically based on the overall team strategy, the individual skill $z^i$ should depend on the team skill $Z$. According to these intuitions, we employ a PGM illustrated in Fig. 2(b) to model the multi-agent skill discovery problem, where the team skill $Z$ is conditioned on the global state $s_t$ and the individual skill $z^i$ is conditioned on both the team skill $Z$ and agent $g^i$'s partial observation $o_t^i$. With the PGM, we formulate multi-agent skill discovery as an inference problem. Then, we perform structured variational inference to derive our objective, which is to optimize a variational lower bound as follows:

$$\log p(\mathcal{O}_{0:T}) \geq \mathbb{E}_{\tau \sim q(\tau)} \Big[ \sum_{t=0}^{T} \Big( r(s_t, \boldsymbol{a}_t) + \underbrace{\log p(Z|s_t) + \sum_{i=1}^{n} \log p(z^i | o_t^i, Z)}_{\text{diversity term}}$$

$$\underbrace{- \log q(Z|s_t) - \sum_{i=1}^{n} \log q(z^i | o_t^i, Z)}_{\text{skill entropy term}} \underbrace{- \sum_{i=1}^{n} \log q(a_t^i | o_t^i, z^i)}_{\text{action entropy term}} \Big) \Big], \tag{3}$$

where $\tau = \big(s_{0:T}, \boldsymbol{o}_{0:T}, \boldsymbol{a}_{0:T}, Z, z^{1:n}\big)$ is the joint trajectory containing states, observations, actions and skills. The detailed derivation of Eq. 3 is shown in Appendix B.

Optimizing this lower bound is to maximize the team reward $r(s_t, \boldsymbol{a}_t)$ and three terms. The diversity term shows the probability of skills on their corresponding states and observations, which can be maximized to encourage different skills to visit different states and observations, and is a crucial element for learning diverse skills. The skill entropy term and the action entropy term reflect the entropy of skills and actions at the visited states and observations, respectively. Optimizing the two entropy terms can enhance the exploration during skill learning.

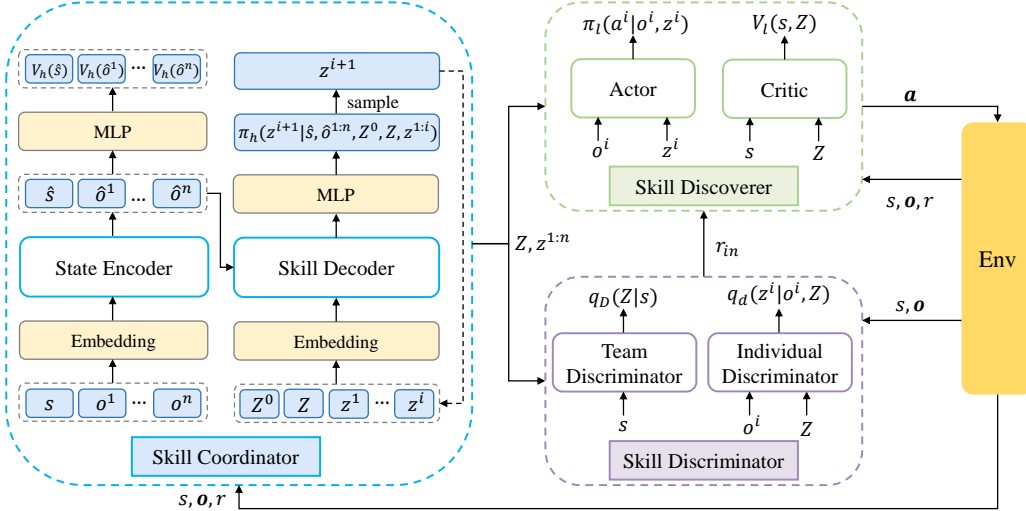

Figure 3: The overall framework of HMASD. At the high level, the skill coordinator adopts a transformer structure to assign team skill $Z$ and individual skills $z^{1:n}$ to agents. At the low level, the skill discoverer chooses primitive action $a^i$ for agent $g^i$ conditioned on the assigned skill $z^i$ and forms a joint action $\boldsymbol{a}$ to interact with the environment. The environment returns global state $s$, joint observation $\boldsymbol{o}$ and team reward $r$. To make the skills diverse and distinguishable, two skill discriminators are employed to generate intrinsic rewards $r_{in}$ for the skill discoverer.

## 3.2 Hierarchical Multi-Agent Skill Discovery

In this subsection, we present Hierarchical Multi-Agent Skill Discovery (HMASD), a practical two-level hierarchical MARL algorithm designed to optimize the derived lower bound in Eq. 3. To estimate the probability distributions in the lower bound, we utilize four approximate functions. Specifically, we first employ a team skill discriminator $q_D(Z|s_t)$ to approximate $p(Z|s_t)$. Then, we use an individual skill discriminator $q_d(z^i|o_t^i, Z)$ to approximate $p(z^i|o_t^i, Z)$. Moreover, a skill coordinator $\pi_h(Z, z^{1:n}|s_t, \boldsymbol{o}_t)$ is utilized to approximate both $q(Z|s_t)$ and $q(z^i|o_t^i, Z)$, while a skill discoverer $\pi_l(a_t^i|o_t^i, z^i)$ is applied to approximate $q(a_t^i|o_t^i, z^i)$. As shown in Fig. 3, these approximate functions are integrated into a two-level hierarchical structure for multi-agent skill discovery. At the high level, the skill coordinator assigns team skill $Z$ and individual skills $z^{1:n}$ to agents every $k$ timesteps, where $k \in \mathbb{N}^+$ is the number of timesteps between two consecutive skill assignments and is called *skill interval*. At the low level, the skill discoverer employs a latent-conditioned policy for each agent to explore the assigned skills using intrinsic rewards generated by the skill discriminators. Below, we describe how to learn these approximate functions for optimizing the derived lower bound.

**Skill Coordinator** Transformer [15] has recently shown great potentials in MARL [27–31]. In this work, we take inspiration from MAT [31], which applies a transformer structure to map the agents' observation sequence into the agents' action sequence. Similar to MAT, we employ a transformer structure for the skill coordinator in our method as shown in Fig. 3. Specifically, we take the sequence of state and observations $(s, o^1, o^2, \cdots, o^n)$ as inputs and embed them into vectors with the same dimension. Then, the embedded sequence is passed through a state encoder, which contains several encoding blocks. Each encoding block consists of a self-attention mechanism, a multi-layer perceptron (MLP) and residual connections. The state encoder outputs a sequence denoted as $(\hat{s}, \hat{o}^1, \hat{o}^2, \cdots, \hat{o}^n)$, which encodes $(s, o^1, o^2, \cdots, o^n)$ into informative representations and is used to approximate the high-level value function $V_h$. After encoding the state and observations, we assign skills in an auto-regressive way. We start with an arbitrary symbol $Z^0$, which is embedded and then fed into a skill decoder containing several decoding blocks. Each decoding block consists of two masked self-attention mechanisms, an MLP and residual connections, where the query of the second masked self-attention mechanism is the output of state encoder. The output of skill decoder is then fed to an MLP to generate the high-level policy $\pi_h(Z|\hat{s}, \hat{o}^{1:n}, Z^0)$. We sample a team skill $Z$ from the policy and insert it back into the decoder to generate $z^1$. By analogy, after $n + 1$ decoding rounds, we get team skill $Z$ and all agents' individual skills $z^{1:n}$. In this way, agent $g^i$'s individual skill $z^i$

depends on team skill $Z$ and all previous agents' individual skills $z^{1:i-1}$, which can prevent skill duplication and allow agents to choose complementary individual skills based on the team skill.

**Skill Discoverer**  After assigning skills sequentially, we learn a shared skill discoverer to explore the assigned skills for agents. The skill discoverer consists of a decentralized actor and a centralized critic as shown in Fig. 3. The decentralized actor takes the partial observation $o^i$ and individual skill $z^i$ as inputs, and outputs the low-level policy $\pi_l(a^i|o^i, z^i)$ that chooses action $a^i$ for each agent $g^i$. All agents share a centralized critic to approximate the low-level value function $V_l(s, Z)$ based on the global state $s$ and team skill $Z$. In our implementation, both the actor and the critic consist of an MLP, a GRU [32] and another MLP. The actor aims to learn the assigned individual skill $z^i$ for each agent $g^i$, while the critic takes a global view and expects to guide the joint behavior of all agents to discover the assigned team skill $Z$.

**Skill Discriminator**  To make the learned skills diverse and distinguishable, we learn two skill discriminators, a team discriminator and an individual discriminator. The team discriminator inputs the global state $s$ and outputs the probability of each team skill $Z \in \mathcal{Z}$, denoted as $q_D(Z|s)$. In addition, we employ an individual discriminator $q_d(z^i|o^i, Z)$, which takes the partial observation $o^i$ and team skill $Z$ as inputs and then outputs the probability of each individual skill $z^i \in \mathcal{X}$. Both discriminators are composed of an MLP. The team discriminator aims to discriminate the team skill based on the global state, while the individual discriminator is designed to discriminate the individual skill given the observation and team skill. The two skill discriminators are utilized to generate intrinsic rewards $\log q_D(Z|s)$ and $\log q_d(z^i|o^i, Z)$, respectively. $\log q_D(Z|s)$ is to reward all agents to jointly explore global states that are easy to discriminate, while $\log q_d(z^i|o^i, Z)$ rewards each agent $g^i$ for visiting those easily distinguishable observations given the team skill $Z$. In other words, it encourages different skills to explore different areas of the state-observation space. Since if two skills explore the same state, this state will be hard to discriminate and then lead to low intrinsic rewards. Therefore, the intrinsic rewards can guide agents to learn diverse and distinguishable skills.

**Overall Training and Execution**  According to Eq. 3, we need to optimize the team reward, the diversity term, the skill entropy term and the action entropy term. At each episode, the high-level skill coordinator assigns skills to agents every $k$ timesteps, then the low-level skill discoverer learns to explore the assigned skills for agents during these $k$ timesteps. We define the sum of team rewards over these $k$ timesteps as the single-step reward for the high-level policy, *i.e.*, the high-level reward can be written as $r_t^h = \sum_{p=0}^{k-1} r_{t+p}$. For the low-level policy, we use a combination of extrinsic team reward and intrinsic rewards, *i.e.*, the low-level reward for agent $g^i$ is:

$$r_t^i = \lambda_e r_t + \lambda_D \log q_D(Z|s_{t+1}) + \lambda_d \log q_d(z^i|o_{t+1}^i, Z), \tag{4}$$

where $\lambda_e$, $\lambda_D$ and $\lambda_d$ are three positive coefficients. The team reward ensures the learned skills are useful for the team performance, while the intrinsic rewards are used to optimize the diversity term in Eq. 3. We adopt the popular PPO [33] objective to optimize both the high-level skill coordinator and the low-level skill discoverer. Given a policy $\pi(a|x)$ with parameters $\theta$, a value function $V(y)$ with parameters $\phi$ and a reward function $r$, we write a generic template for PPO objective as:

$$\mathcal{L}_{\text{PPO}}\{\pi(a|x), V(y), r\} = \mathcal{L}(\theta) + \lambda_c \mathcal{L}(\phi),$$

$$\mathcal{L}(\theta) = -\mathbb{E}_t \left[ \min \left( \frac{\pi(a_t|x_t)}{\pi_{old}(a_t|x_t)} \hat{A}_t, \text{clip} \left( \frac{\pi(a_t|x_t)}{\pi_{old}(a_t|x_t)}, 1 - \epsilon, 1 + \epsilon \right) \hat{A}_t \right) \right], \tag{5}$$

$$\mathcal{L}(\phi) = \mathbb{E}_t \left[ \max \left\{ \left( V(y_t) - \hat{R}_t \right)^2, \left( V_{old}(y_t) + \text{clip} \left( V(y_t) - V_{old}(y_t), -\epsilon, \epsilon \right) - \hat{R}_t \right)^2 \right\} \right],$$

where $\hat{A}_t$ is the advantage computed using GAE [34], $\hat{R}_t = \hat{A}_t + V_{old}(y_t)$ and $\lambda_c$ is the coefficient of value loss. Then, we write the overall objective for the high-level skill coordinator as:

$$\mathcal{L}_h(\theta_h, \phi_h) = \mathcal{L}_{\text{PPO}}\{\pi_h(Z|\hat{s}, \hat{o}^{1:n}, Z^0), V_h(\hat{s}), r_t^h\} + \sum_{i=1}^{n} \mathcal{L}_{\text{PPO}}\{\pi_h(z^i|\hat{s}, \hat{o}^{1:n}, Z^0, Z, z^{1:i-1}), V_h(\hat{o}^i), r_t^h\}$$

$$- \lambda_h \left( \mathbb{E} \left[ \mathcal{H}(\pi_h(Z|\hat{s}, \hat{o}^{1:n}, Z^0)) \right] + \sum_{i=1}^{n} \mathbb{E} \left[ \mathcal{H}(\pi_h(z^i|\hat{s}, \hat{o}^{1:n}, Z^0, Z, z^{1:i-1})) \right] \right), \tag{6}$$

where $\mathcal{H}(\pi_h(\cdot))$ is the entropy of the high-level policy that aims to optimize the skill entropy term in Eq. 3, and $\lambda_h$ is the high-level entropy coefficient. $\theta_h$ and $\phi_h$ denote the parameters of the policy and

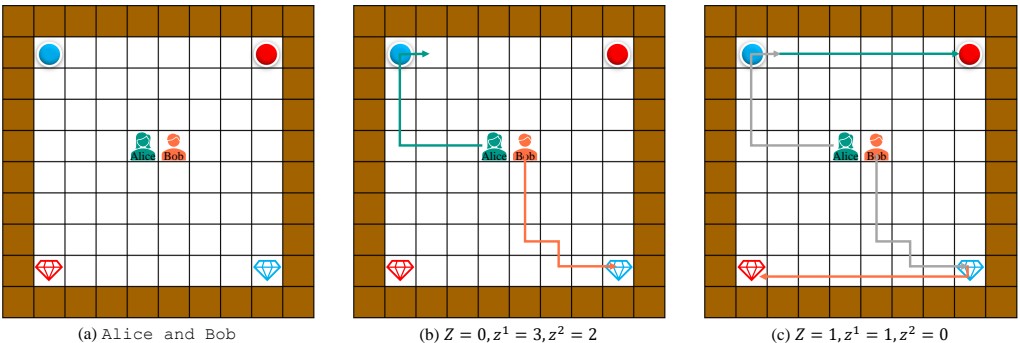

(a) `Alice_and_Bob`  (b) $Z = 0, z^1 = 3, z^2 = 2$  (c) $Z = 1, z^1 = 1, z^2 = 0$

Figure 4: (a) The `Alice_and_Bob` game. The objective is to collect both diamonds, where each diamond is allowed to be collected only when an agent is standing on the button with the same color. (b) At timestep $t = 0$, HMASD selects the team skill of collecting the blue diamond ($Z = 0$) and two individual skills for reaching the blue button ($z^1 = 3$) and blue diamond ($z^2 = 2$). (c) After collecting the blue diamond, the team skill transitions to collecting the red diamond ($Z = 1$) with two individual skills of reaching the red button ($z^1 = 1$) and red diamond ($z^2 = 0$) at timestep $t = k$, where $k$ is the skill interval.

value function in the skill coordinator, respectively. Similarly, the overall objective for the low-level skill discoverer is:

$$\mathcal{L}_l(\theta_l, \phi_l) = \sum_{i=1}^{n} \mathcal{L}_{\text{PPO}}\{\pi_l(a^i|o^i, z^i), V_l(s, Z), r_t^i\} - \lambda_l \sum_{i=1}^{n} \mathbb{E}\left[\mathcal{H}(\pi_l(a^i|o^i, z^i))\right], \qquad (7)$$

where $\mathcal{H}(\pi_l(\cdot))$ denotes low-level policy entropy and optimizes the action entropy term in Eq. 3, and $\lambda_l$ is the low-level entropy coefficient. $\theta_l$ and $\phi_l$ represent the parameters of the low-level policy and value function, respectively. The skill discriminator is trained in a supervised manner with the categorical cross-entropy loss:

$$\mathcal{L}_d(\phi_D, \phi_d) = -\mathbb{E}_{(s,Z)\sim\mathcal{D}}\left[\log q_D(Z|s)\right] - \sum_{i=1}^{n}\mathbb{E}_{(o^i,Z,z^i)\sim\mathcal{D}}\left[\log q_d(z^i|o^i, Z)\right], \qquad (8)$$

where $\mathcal{D} = \{(s, Z, \boldsymbol{o}, z^{1:n})\}$ is a dataset storing the state-skill pairs during training. $\phi_D$ and $\phi_d$ are the parameters of team discriminator and individual discriminator, respectively. The pseudo code of our method is shown in Appendix A.

During the execution phase, we only use the centralized high-level policy $\pi_h(Z, z^{1:n}|s, \boldsymbol{o})$ and the decentralized low-level policy $\pi_l(a^i|o^i, z^i)$. For every $k$ timesteps, the high-level policy $\pi_h(Z, z^{1:n}|s, \boldsymbol{o})$ first chooses a team skill $Z$ (*i.e.*, team strategy) from a global perspective and then sequentially assigns complementary individual skills $z^{1:n}$ to agents based on the team strategy $Z$. With the assigned individual skill $z^i$, each agent $g^i$ selects an action $a^i$ according to the low-level policy $\pi_l(a^i|o^i, z^i)$ at every timestep for execution. Therefore, our method performs one timestep of centralized execution and $k - 1$ timesteps of decentralized execution in every $k$ timesteps. Such periodic and spaced centralized execution can coordinate agents more efficiently from a global view compared to the fully decentralized execution, and a small amount of centralized execution is acceptable in many multi-agent tasks. For example, during a basketball game, the coach can call a timeout and gather all players to adjust the team strategy and each player's individual strategy. In short, our method achieves a balance between fully centralized execution [31, 35] and fully decentralized execution [4, 36–39].

## 4   Experiments

In this section, we evaluate the effectiveness of our method. We first conduct a case study to show how HMASD effectively learns diverse useful skills and combines them to complete the task. Then, we compare HMASD with strong MARL baselines on two challenging sparse reward multi-agent benchmarks, *i.e.*, SMAC [40] with 0-1 reward and Overcooked [41]. We further perform ablation studies for HMASD to confirm the benefits of components in our method. We select MAPPO [5], MAT [31] and MASER [42] as our baselines. MAPPO and MAT are two strong policy-based MARL algorithms that achieve state-of-art performance on various multi-agent tasks [40, 43–45]. MASER

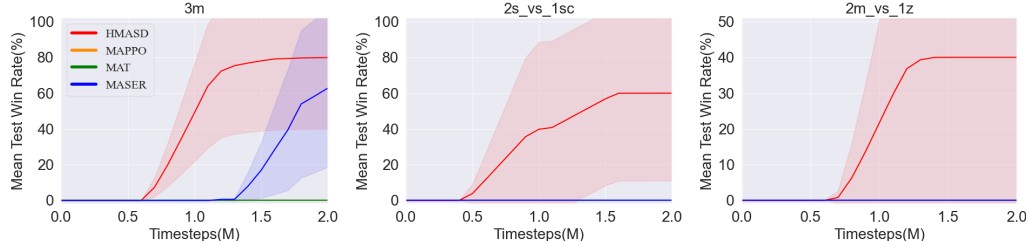

Figure 6: Performance comparison between HMASD and baselines on SMAC with 0-1 reward.

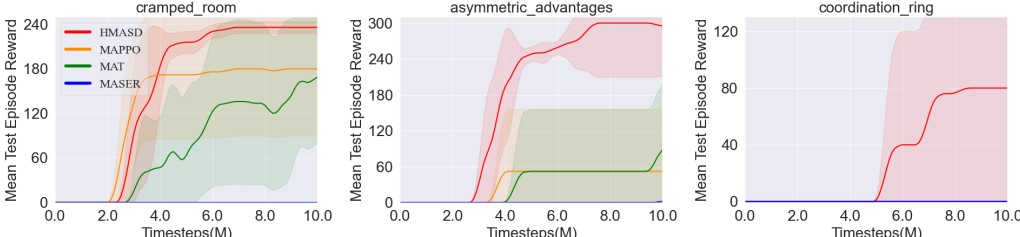

Figure 7: Performance comparison between HMASD and baselines on Overcooked.

presents a goal-conditioned method for solving sparse reward MARL. For all methods, we show the mean and variance of the performance across five different random seeds. The hyperparameter setting can be found in Appendix E.

### 4.1 Case Study

In this subsection, we design a toy game `Alice_and_Bob` to demonstrate how our method works. As shown in Fig. 4(a), the `Alice_and_Bob` game is an $8 \times 8$ grid world environment surrounded by walls. There are two agents Alice and Bob with random initial positions, two buttons at the top and two diamonds at the bottom. The goal of the game is to collect both diamonds, where each diamond is allowed to be collected only when an agent is standing on the button with the same color. Alice and Bob can receive a non-zero team reward only after they cooperatively collect both diamonds. We set the number of team skills $n_Z = 2$ and the number of individual skills $n_z = 4$ for HMASD. Fig. 4(b) and (c) show the learned skills of our method. We can observe that the two team skills $Z = 0, 1$ correspond to collecting the blue diamond and red diamond for the whole team, respectively, and the four individual skills $z^i = 0, 1, 2, 3$ guide the individual agent to reach the red diamond, red button, blue diamond and blue button, respectively. We observe similar skills for agents with different initial positions in Appendix C. By learning these diverse and useful skills, our method achieves a higher success rate of task completion than baselines as shown in Fig. 5. These results demonstrate that HMASD could discover significant team and individual skills, and effectively combine them to accomplish the task.

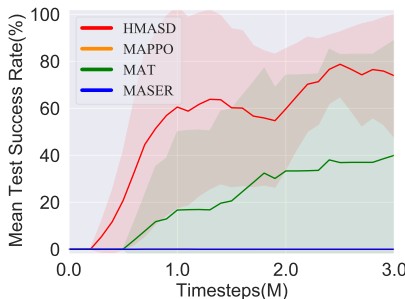

Figure 5: Performance comparison between HMASD and baselines on `Alice_and_Bob`.

### 4.2 Performance on Sparse Reward Multi-Agent Benchmarks

In this subsection, we first test our method on a widely-used MARL benchmark, SMAC [40], where we learn to control a team of ally units against a team of enemy units controlled by a built-in strategy. The ally team wins the game only if all enemy units are killed within the time limit of an episode. Our objective is to maximize the win rate for the ally team. The default reward setting of SMAC contains many dense rewards, such as the unit's health and damage. These dense rewards enable many MARL algorithms like MAT [31] and MAPPO [5] to attain almost 100% win rate on all scenarios. However, designing useful dense rewards is usually expensive and adjusting the weights between rewards is

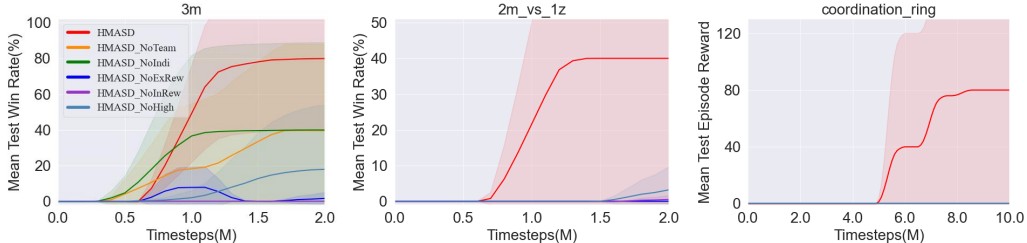

Figure 8: Ablation studies regarding components of HMASD.

time-consuming. In this work, we consider a more general reward setting, *i.e.*, SMAC with 0-1 reward, which returns a non-zero team reward of 1 only at the last timestep if the ally team wins. As shown in Fig. 6, HMASD significantly outperforms baselines on SMAC with 0-1 reward, where MAT and MAPPO don't work at all. MASER only works on the scenario 3m and fails on the other two scenarios 2s_vs_1sc and 2m_vs_1z. This demonstrates the effectiveness of HMASD and the high efficiency of skill discovery for solving sparse reward multi-agent tasks.

Next, we evaluate HMASD on another sparse reward multi-agent task called Overcooked, a popular cooperative cooking simulation game. We follow a simplified version of the game proposed in [41], the objective of which is to deliver the soup as fast as possible. Each soup requires agents to place 3 onions in a pot, cook them for 20 timesteps, put the cooked soup in a dish, and deliver it to the service desk. All agents will receive a team reward of 20 after delivering a soup. In Fig. 7, we compare HMASD with baselines on three Overcooked scenarios. It can be observed that HMASD could also achieve a superior performance over baselines on Overcooked, which further confirms that HMASD can discover useful skills to effectively complete the task.

Moreover, we visualize the learned skills and find that HMASD uses only a few skills to complete the complex task after training. This is because HMASD will encourage different skills to explore different state-observation spaces, but only a small part of the state-observation space can result in a non-zero team reward in the complex sparse reward multi-agent task. For example, on the SMAC scenario, there is usually a big map, we observe that most of the learned skills explore areas of the map without enemies and thus don't contribute to the team reward. These results indicate that when the state-observation space is large, HMASD can discover diverse skills but maybe only some of them are useful for the team reward. More fine-grained results can be found in Appendix F.

Additionally, we compare HMASD with an exploration bonus MARL baseline [46] in Appendix H. The results show that the performance improvements of HMASD mainly come from diverse skill discovery and effective skill combination, rather than implicit exploration.

## 4.3 Ablation Studies

In this subsection, we conduct ablation studies to investigate the impact of three main components in HMASD: (1) discovering both team and individual skills, (2) using a combination of extrinsic team reward and intrinsic rewards for the low-level reward and (3) employing the high-level skill coordinator to assign skills to agents. To test component (1), we introduce two variants of HMASD, denoted as HMASD_NoTeam and HMASD_NoIndi, respectively. HMASD_NoTeam only learns individual skills for agents, while HMASD_NoIndi only lets all agents learn a shared team skill. To evaluate component (2), we consider HMASD_NoExRew and HMASD_NoInRew, which set $\lambda_e = 0$ and $\lambda_D = \lambda_d = 0$ in Eq. 4, respectively. As for component (3), we adopt HMASD_NoHigh, which removes the high-level policy and randomly assigns skills to agents at the start of each episode. As shown in Fig. 8, HMASD performs better than both HMASD_NoTeam and HMASD_NoIndi, which emphasizes the importance of discovering both team and individual skills in multi-agent tasks. After removing the extrinsic team reward or intrinsic rewards in the low-level reward, the performance of HMASD has a large drop. Especially without intrinsic rewards, HMASD can't work on most scenarios. This highlights the considerable contributions of optimizing the diversity term in Eq. 3. Besides, the performance comparison between HMASD and HMASD_NoHigh reveals the necessity of the high-level policy. More ablations for HMASD are shown in Appendix D.

# 5   Related Work

**Skill Discovery in MARL**   Skill discovery is a promising approach to solving complex tasks for its ability to discover diverse skills even without any extrinsic reward. Recently, this approach has been extended to MARL. MASD [9] learns coordinating skills adversarially by setting an information bottleneck. VMAPD [10] constructs a diverse Dec-POMDP to learn diverse skills as different solutions for multi-agent tasks. However, these two methods focus on learning a shared team skill for all agents, and use only one skill throughout an episode. To combine different skills to tackle multi-agent tasks, HSD [11] employs a hierarchical structure, where the high-level policy selects low-level skills for agents based on local observations. HSL [12] follows a similar hierarchical structure and introduces a skill representation mechanism to enhance skill learning. Nevertheless, these two methods only learn individual skills from each agent's individual perspective, without considering team skill learning. Besides, in these two methods, each agent selects skills only based on its local observation, leading to potential skill duplication between agents with similar observations. ODIS [47] applies skill discovery to offline MARL, which discovers generalizable individual skills across different tasks from offline multi-task data, and trains a coordination policy to assign skills to agents with the centralized training and decentralized execution paradigm [4]. However, this decentralized skill assignment may also lead to skill duplication. Chen et al. [48, 49] learn joint options (*i.e.*, skills) by approximating the joint state space as the Kronecker product of the state spaces of individual agents. In this work, we propose to discover both team skills and individual skills. Moreover, we assign skills to agents from a global perspective to enable complementary skill selection and prevent skill duplication effectively.

**Sparse Reward MARL**   The issue of sparse reward poses a significant obstacle when applying RL to solve real-world problems, which will be exacerbated in multi-agent tasks. Several approaches have been proposed to address the sparse reward problem in MARL. SEAC [50] leverages experience sharing among agents to enable efficient exploration. CMAE [51] promotes cooperative exploration by selecting a shared goal for agents from state spaces. VACL [52] tackles the sparse reward problem in MARL by utilizing an automatic curriculum learning algorithm that incrementally expands the training tasks from easy to hard. More recently, MASER [42] introduces a goal-conditioned method that generates subgoals for agents from the experience replay buffer. In this work, we present a hierarchical MARL algorithm that discovers underlying skills and effectively combines these skills to solve sparse reward multi-agent tasks.

# 6   Conclusion

It is an efficient way to learn a set of skills and combine them properly to tackle complex tasks. In this study, we take advantage of skill discovery to address MARL problems. We propose to discover latent team and individual skills by embedding them into a probabilistic graphical model. In this way, we formulate multi-agent skill discovery as an inference problem, and derive a variational lower bound as the optimization objective. We then design a practical MARL method called HMASD to optimize the lower bound, where different team skills explore different global state spaces and different individual skills explore different local observation spaces. The empirical results show that HMASD significantly improves performance on challenging sparse reward multi-agent tasks by learning diverse team and individual skills with efficient skill combination.

## Acknowledgments

This work is supported by National Key R&D Program of China under Contract 2022ZD0119802, and National Natural Science Foundation of China under Contract 61836011. It was also supported by GPU cluster built by MCC Lab of Information Science and Technology Institution, USTC, and the Supercomputing Center of the USTC. What's more, it is funded by Collective Intelligence & Collaboration Laboratory (Open Fund Project No. QXZ23014101) and by Young Elite Scientists Sponsorship Program by CAST 2022QNRC001.

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

# A Pseudo Code of Hierarchical Multi-Agent Skill Discovery

---

**Algorithm 1:** Hierarchical Multi-Agent Skill Discovery

---

**Initialize**: Skill coordinator parameters $\theta_h$ and $\phi_h$, Skill discoverer parameters $\theta_l$ and $\phi_l$, Skill discriminator parameters $\phi_D$ and $\phi_d$, High-level replay buffer $\mathcal{B}_h$, Low-level replay buffer $\mathcal{B}_l$, State-skill pair dataset $\mathcal{D}$.

**for** $episode = 0, 1, \cdots, K$ **do**
    # The inference phase
    **for** $t = 0, 1, \cdots, T$ **do**
      Collect global state $s_t$ and agents' partial observations $\mathbf{o}_t = (o_t^1, \cdots, o_t^n)$ from environments.
      **if** $t \mod k = 0$ **then**
        | Sample skills $Z, z^{1:n} \sim \pi_h(Z, z^{1:n}|s_t, \mathbf{o}_t)$.
      **end**
      **for** *each agent* $g^i \in \mathcal{N}$ **do**
        | Sample action $a_t^i \sim \pi_l(a_t^i|o_t^i, z^i)$.
      **end**
      Execute joint actions $\boldsymbol{a}_t = (a_t^1, \cdots, a_t^n)$ in environments and collect the team reward $r_t$, next state $s_{t+1}$ and next observations $\mathbf{o}_{t+1}$.
      **for** *each agent* $g^i \in \mathcal{N}$ **do**
        | Compute low-level reward $r_t^i = \lambda_e r_t + \lambda_D \log q_D(Z|s_{t+1}) + \lambda_d \log q_d(z^i|o_{t+1}^i, Z)$.
        | Store $(s_t, Z, o_t^i, z^i, a_t^i, r_t^i)$ into $\mathcal{B}_l$.
      **end**
      **if** $t \mod k = k - 1$ **then**
        | Store $\left(s_{t-k-1}, \mathbf{o}_{t-k-1}, Z, z^{1:n}, \sum_{p=0}^{k-1} r_{t-p}\right)$ into $\mathcal{B}_h$.
      **end**
      Store $(s_{t+1}, Z, \mathbf{o}_{t+1}, z^{1:n})$ into $\mathcal{D}$.
    **end**
    # The training phase
    Sample a random minibatch of data from $\mathcal{B}_h$ to update $\theta_h$ and $\phi_h$ by minimizing $\mathcal{L}_h(\theta_h, \phi_h)$.
    Sample a random minibatch of data from $\mathcal{B}_l$ to update $\theta_l$ and $\phi_l$ by minimizing $\mathcal{L}_l(\theta_l, \phi_l)$.
    Sample a random minibatch of data from $\mathcal{D}$ to update $\phi_D$ and $\phi_d$ by minimizing $\mathcal{L}_d(\phi_D, \phi_d)$.
**end**

---

# B Derivation of the Variational Lower Bound

In this section, we adopt the structured variational inference to formulate our objective based on the PGM in Fig. 2(b). In structured variational inference, approximate inference is performed by optimizing the variational lower bound [14]. Let $\tau = (s_{0:T}, \boldsymbol{o}_{0:T}, \boldsymbol{a}_{0:T}, Z, z^{1:n})$, the variational lower bound is given by:

$$
\begin{aligned}
\log p(\mathcal{O}_{0:T}) &= \log \int p(\mathcal{O}_{0:T}, \tau) d\tau \\
&= \log \int p(\mathcal{O}_{0:T}, \tau) \frac{q(\tau)}{q(\tau)} d\tau \\
&= \log \mathbb{E}_{\tau \sim q(\tau)} \left[ \frac{p(\mathcal{O}_{0:T}, \tau)}{q(\tau)} \right] \\
&\geq \mathbb{E}_{\tau \sim q(\tau)} \left[ \log p(\mathcal{O}_{0:T}, \tau) - \log q(\tau) \right],
\end{aligned}
$$

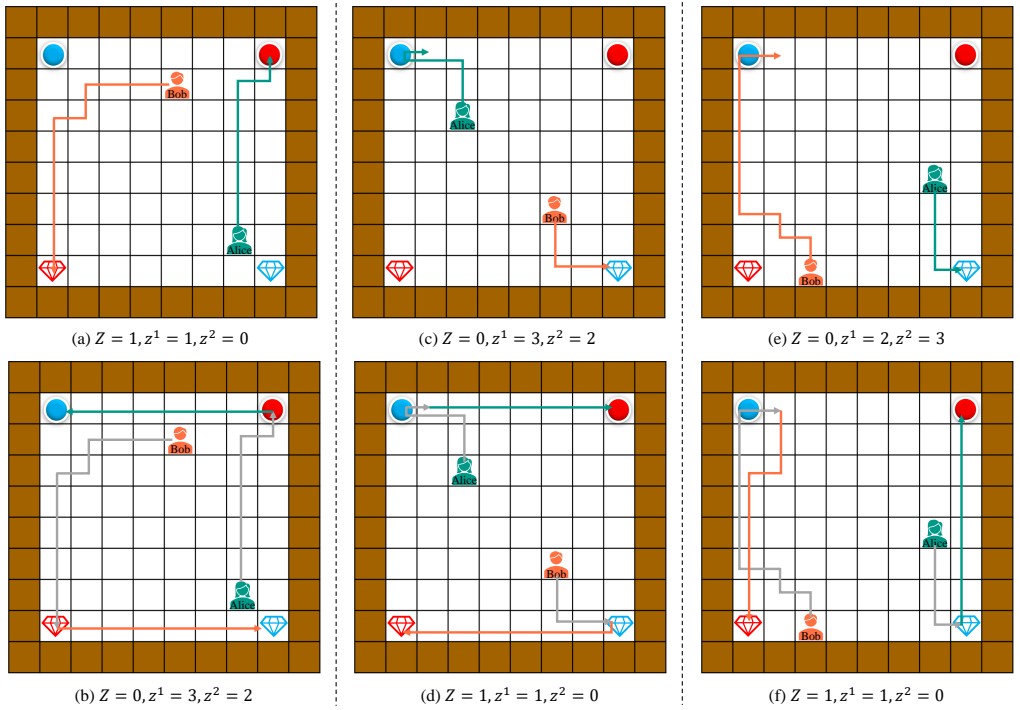

(a) $Z = 1, z^1 = 1, z^2 = 0$     (c) $Z = 0, z^1 = 3, z^2 = 2$     (e) $Z = 0, z^1 = 2, z^2 = 3$

(b) $Z = 0, z^1 = 3, z^2 = 2$     (d) $Z = 1, z^1 = 1, z^2 = 0$     (f) $Z = 1, z^1 = 1, z^2 = 0$

Figure 9: Visualization of learned team and individual skills on `Alice_and_Bob`. (a-b), (c-d), (e-f) show the skills learned by the agents with three different initial positions, respectively.

where the inequality in the last line is derived using Jensen's inequality. We assume that the action prior distribution $p(\boldsymbol{a}_t)$ is a uniform distribution. Then, we have:

$$p(\mathcal{O}_{0:T}, \tau) = p(s_0) \prod_{t=0}^{T} p(\boldsymbol{o}_t) p(s_{t+1}|s_t, \boldsymbol{a}_t) p(\mathcal{O}_t|s_t, \boldsymbol{a}_t) p(Z|s_t) p(z^1|o_t^1, Z) \cdots p(z^n|o_t^n, Z)$$

$$q(\tau) = q(s_0) \prod_{t=0}^{T} q(\boldsymbol{o}_t) q(s_{t+1}|s_t, \boldsymbol{a}_t) q(Z|s_t) q(z^1|o_t^1, Z) \cdots q(z^n|o_t^n, Z) q(a_t^1|o_t^1, z^1) \cdots q(a_t^n|o_t^n, z^n).$$

Similar to [14], we fix $q(s_0) = p(s_0), q(\boldsymbol{o}_t) = p(\boldsymbol{o}_t), q(s_{t+1}|s_t, \boldsymbol{a}_t) = p(s_{t+1}|s_t, \boldsymbol{a}_t)$ and let $p(\mathcal{O}_t|s_t, \boldsymbol{a}_t) = \exp(r(s_t, \boldsymbol{a}_t))$. The bound reduces to:

$$\log p(\mathcal{O}_{0:T}) \geq \mathbb{E}_{\tau \sim q(\tau)} \left[ \log p(\mathcal{O}_{0:T}, \tau) - \log q(\tau) \right]$$

$$= \mathbb{E}_{\tau \sim q(\tau)} \Big[ \sum_{t=0}^{T} \Big( r(s_t, \boldsymbol{a}_t) + \log p(Z|s_t) + \sum_{i=1}^{n} \log p(z^i|o_t^i, Z)$$

$$- \log q(Z|s_t) - \sum_{i=1}^{n} \log q(z^i|o_t^i, Z) - \sum_{i=1}^{n} \log q(a_t^i|o_t^i, z^i) \Big) \Big].$$

## C   Visualization of skills on `Alice_and_Bob`

We visualize the learned team and individual skills on `Alice_and_Bob` by the agents with different initial positions in Fig. 9. With different initial positions, agents can learn similar team and individual skills. Specifically, team skills $Z = 0$ and $Z = 1$ guide the whole team to collect the blue diamond and the red diamond, respectively. Individual skills $z^i = 0$, $z^i = 1$, $z^i = 2$ and $z^i = 3$ guide the individual agent to reach the red diamond, red button, blue diamond and blue button, respectively.

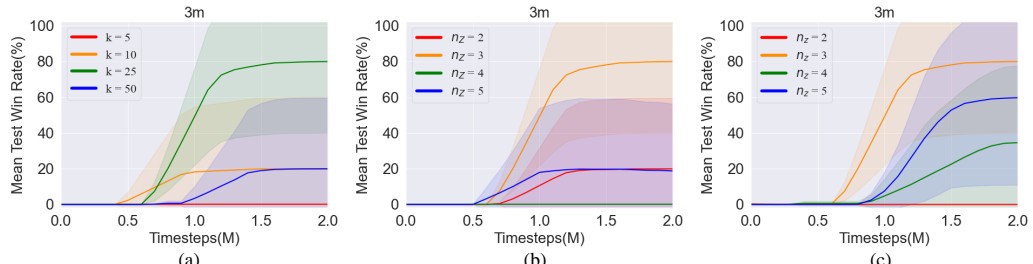

Figure 10: More ablations for HMASD. (a) The performance of HMASD with different skill intervals. (b) The performance of HMASD with different numbers of team skills. (c) The performance of HMASD with different numbers of individual skills.

## D More Ablations

We conduct more ablations about the skill interval $k$, the number of team skills $n_Z$ and the number of individual skills $n_z$ in Fig. 10. Results reveal that HMASD performs poorly with too short or too long skill intervals, which demonstrates that an appropriate number of timesteps is necessary for each skill to be learned effectively. Furthermore, it can be observed that the performance of HMASD varies greatly with different numbers of team skills and individual skills, which shows that HMASD is not very stable on sparse reward multi-agent tasks. The limitations of our work are discussed in Appendix G.

## E Hyperparameter Setting

For all baselines, we use the open source code of the original paper. We implement HMASD based on the codebase of MAT and MAPPO. The hyperparameters for different tasks are presented in Table 1- 3. In particular, we want to emphasize the setting of $\lambda_e$, *i.e.*, the weight of team reward in the low-level reward. It guides the skills to be useful for the team performance. On the simple `Alice_and_Bob`, the state-observation space is small, and it's easy for skills to explore those states that contribute to the team reward. So we set $\lambda_e = 0$. On the complex SMAC with 0-1 reward and Overcooked, the state-observation space is large, and it's very difficult to explore those rare states that induce positive team reward. We set $\lambda_e = 100$, which is much larger than the weights of intrinsic rewards $\lambda_D$ and $\lambda_d$. In this way, the team reward can dominate the low-level reward when agents encounter positive team reward, which guides the agents to learn to complete the task more quickly.

Table 1: Common hyperparameters used for HMASD, MAT and MAPPO across all tasks.

| hyperparameters | value | hyperparameters | value | hyperparameters | value |
|---|---|---|---|---|---|
| training threads | 16 | rollout threads | 32 | hidden size | 64 |
| use valuenorm | True | use orthogonal | True | gain | 0.01 |
| optimizer | Adam | optimizer epsilon | 1e-5 | weight decay | 0 |
| ppo epoch | 15 | clip param | 0.2 | num mini batch | 1 |
| value loss coef | 1 | use gae | True | gae lambda | 0.95 |
| gamma | 0.99 | use huber loss | True | huber delta | 10 |

Table 2: Common hyperparameters used for HMASD, MAT and MAPPO in different tasks.

| task | lr | episode length | num env steps | eval episodes | eval rollout threads |
|---|---|---|---|---|---|
| `Alice_and_Bob` | 5e-4 | 100 | 3e6 | 100 | 1 |
| SMAC with 0-1 reward | 1e-4 | 100 | 2e6 | 100 | 4 |
| Overcooked | 1e-4 | 400 | 1e7 | 32 | 8 |

Table 3: Different hyperparameters used for HMASD in different scenarios.

| scenario | $k$ | $n_Z$ | $n_z$ | $\lambda_h$ | $\lambda_l$ | $\lambda_e$ | $\lambda_D$ | $\lambda_d$ |
|---|---|---|---|---|---|---|---|---|
| Alice_and_Bob | 50 | 2 | 4 | 0.1 | 0.01 | 0 | 0.1 | 0.2 |
| 3m | 25 | 3 | 3 | 0.001 | 0.01 | 100 | 0.1 | 0.5 |
| 2s_and_1sc | 25 | 2 | 5 | 0.03 | 0.01 | 100 | 0.1 | 1 |
| 2m_and_1z | 50 | 3 | 2 | 0.005 | 0.01 | 100 | 0.1 | 1 |
| cramped_room | 25 | 3 | 3 | 0.01 | 0.01 | 100 | 1 | 0.5 |
| asymmetric_advantages | 25 | 3 | 3 | 0.01 | 0.01 | 100 | 0.1 | 0.1 |
| coordination_ring | 10 | 3 | 3 | 0.1 | 0.01 | 100 | 0.15 | 0.1 |

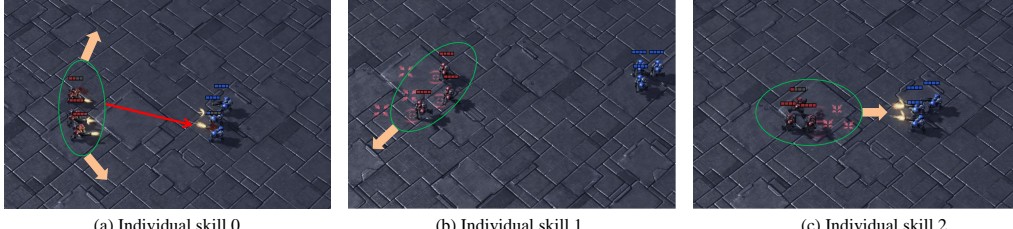

(a) Individual skill 0     (b) Individual skill 1     (c) Individual skill 2

Figure 11: Visualizations of 3 learned individual skills on the SMAC scenario 3m. To visualize the $i^{th}$ individual skill, we set all agents' individual skills to be $i$, *i.e.*, $z^1 = z^2 = z^3 = i$, where $i = 0, 1, \cdots, n_z - 1$. (a) For individual skill 0, agents learn to spread out to both sides to form an arc formation and focus fire on the same enemy at the same time, which can quickly reduce the number of enemies and win the game. (b) For individual skill 1, agents learn to explore the bottom left area of the map, where there are no enemies. In the end, agents lose the game for not killing all enemies within the time limit. (c) For individual skill 2, agents learn to move right to approach enemies but not attack them in time, which leads to losing the game.

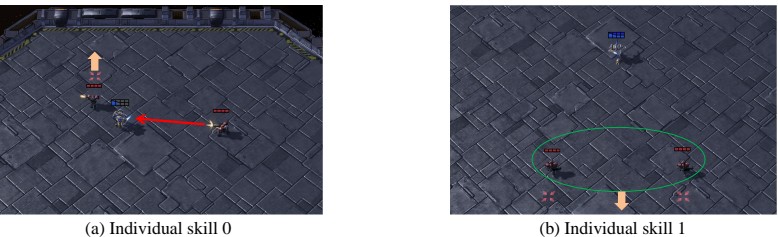

(a) Individual skill 0        (b) Individual skill 1

Figure 12: Visualizations of 2 learned individual skills on the SMAC scenario 2m_vs_1z. (a) For individual skill 0, agents learn to take advantage of attack range to kite the enemy. The agent closer to the enemy will run away, and the other agent will attack the enemy. In this kiting way, the two agents take turns attacking the enemy and ultimately win the game. (b) For individual skill 1, agents learn to explore the bottom area of the map, while the enemy is in the upper area. Finally, agents fail to find the enemy and lose the game.

## F More Fine-grained Results

**Percentage of skills useful for the team reward** We count the average percentage of useful individual skills among all learned individual skills on the SMAC. For each SMAC scenario, we conduct 5 different runs with different random seeds. We learn $3, 5, 2$ individual skills for each run on 3m, 2s_vs_1sc, 2m_vs_1z, respectively. Therefore, we learn $3 * 5 + 5 * 5 + 2 * 5 = 50$ individual skills on three SMAC scenarios. After our test, only 12 individual skills are useful for the team performance. In other words, only $24\%$ of the learned individual skills are useful for completing the task on average after training.

**Percentage of runs that learn meaningful behavior** For every scenario, we conduct 5 runs. For each run on the SMAC scenario, once agents discover useful skills, the performance will quickly

increase to 1. So the final performance is either 1 or 0, which leads to a large variance among 5 runs. There are 7 scenarios in this work. Among $5 * 7 = 35$ runs, 26 runs could learn meaningful behavior. Therefore, the percentage of the model to learn meaningful behavior among all trials is $26/35 = 74.3\%$. One of our future goals is to increase this percentage and reduce the variances of our method.

**Visualization of the learned skills**    We visualize the learned individual skills on SMAC as shown in Fig. 11 and Fig. 12. We can observe that only one skill can result in a non-zero team reward on both scenarios, and the remaining skills explore the state-observation spaces that don't contribute to the team reward. As for the learned skills on Overcooked, we find that in those high-performance runs, all skills are useful and each of them can perform well individually. This may be because Overcooked is a grid world task and has a smaller state-observation space than SMAC. Thus those states with positive rewards are easier to be explored by all skills. These visualizations reflect the potential inefficiency of HMASD in solving complex tasks with large state-observation space.

## G    Limitations

In this section, we discuss three potential limitations of HMASD. Firstly, when the state-observation space is large, HMASD can discover diverse skills but maybe only some of them (about 24% on SMAC) are useful for the team reward. There are a large percentage of skills are assigned to explore those zero-reward states. Secondly, the number of team skills and individual skills should be carefully adjusted. When applying HMASD to a new task, it requires to adjust several hyperparameters as shown in Table 3. Thirdly, HMASD can only learn team skills for the entire team, which lacks flexibility in multi-agent tasks that require team skills within sub-teams. Our future work aims to improve these limitations and make HMASD better at solving sparse-reward multi-agent tasks.

## H    Comparison of HMASD with the Multi-Agent Exploration Baseline

In this section, we compare HMASD with an exploration bonus MARL baseline, EITI/EDTI [46], on Overcooked. EITI/EDTI proposes exploration strategies where agents start with decentralized exploration driven by their individual curiosity (*i.e.*, individual exploration), and are also encouraged to coordinate their exploration (*i.e.*, joint exploration). The final reward for each agent is the sum of team reward, individual exploration bonus and joint exploration bonus. EITI and EDTI provide 2 ways to calculate the joint exploration bonus. The comparison of HMASD with EITI/EDTI on Overcooked is shown in Table 4.

We can see that HMASD outperforms EITI and EDTI on all Overcooked scenarios. Although EITI/EDTI encourages both individual exploration and joint exploration, it doesn't formulate the relationship between individual exploration and joint exploration. These two explorations may have conflicts, causing neither exploration to work well. In our work, we propose to discover both team and individual skills for solving sparse-reward multi-agent tasks. Importantly, we build a probabilistic graphical model to formulate the relationship between team skill and individual skill, and then derive a lower bound as our objective. The results show that our method could discover significant team and individual skills, and effectively combine them to accomplish the sparse-reward multi-agent task.

Table 4: The final episode reward of HMASD and EITI/EDTI on Overcooked.

| Scenario | EITI | EDTI | HMASD |
|---|---|---|---|
| cramped_room | $0.122 \pm 0.013$ | $0.119 \pm 0.015$ | $\mathbf{236.0 \pm 8.0}$ |
| asymmetric_advantages | $0.008 \pm 0.002$ | $0.004 \pm 0.002$ | $\mathbf{295.1 \pm 84.5}$ |
| coordination_ring | $0.0 \pm 0.0$ | $0.0 \pm 0.0$ | $\mathbf{80.0 \pm 97.9}$ |

