# OpenReview forum: "Hierarchical Multi-Agent Skill Discovery"
_NeurIPS.cc/2023/Conference — NeurIPS 2023 poster_

### Official Review · Reviewer_Ty5M · 2023-06-16

**Soundness:** 3 good
**Presentation:** 3 good
**Contribution:** 3 good
**Rating:** 6
**Confidence:** 4

**Summary:**

This paper introduces a framework that concurrently learns the individual skills for each agent and the team skill for the entire team, amalgamating these skills to perform multi-agent tasks. The discovery of skills is grounded in a probabilistic graphical model and employs variational inference tools for scalable optimization, extending unsupervised skill discovery in single-agent RL. The proposed algorithm exhibits superior performance on sparse reward multi-agent benchmarks when compared to robust MARL baselines.

**Strengths:**

(a) The paper is effectively articulated, with clear exposition of the intuitions underlying each aspect of the algorithm design.

(b) The empirical results provide substantial support for the paper's technical contributions.

(c) The extension of unsupervised skill discovery from single-agent to multi-agent RL is a considerable achievement, offering a promising direction for future research.

**Weaknesses:**

(a) The algorithm might face limitations in practical applications due to: (1) the high-level policy necessitating input that comprises the global state and all agents' observations, and (2) the multi-agent options designed for the entire team, which lack flexibility as coordination among agents often manifests within sub-teams.

(b) The algorithm framework is complex, encompassing multiple components and hyperparameters. The fine-tuning efforts, particularly for coordinating the training of neural networks across different scenarios, could be resource-intensive.

(c) The paper omits important details in certain areas and requires more clarity. My queries and suggestions follow.

**Questions:**

(a) The statement in Line 40, "One is to let all agents learn a shared team skill [9, 10], which promotes team cooperation behaviors but suffers from high complexity," lacks clarity. Could you expand on why the complexity can be high? Since your work also aims to discover multi-agent skills, how have you addressed this complexity issue compared to [9, 10]?

(b) There are additional works on multi-agent option (also known as skill) discovery, such as [1]. Please consider offering a comprehensive review of this research field to highlight your novel contributions.

(c) Based on the definition in Line 144, it seems that the agents in the environment must be homogeneous as they share the same individual skill space. Is that correct?

(d) In Line 197, it's unclear how the protocol design can prevent skill duplication.

(e) It's unclear how you can maximize the lower bound in Eq. 3 with the objectives in Eq. 4-8, given that several coefficients are introduced and the optimization of functions are separated.

(f) For fair comparisons, MAPPO, MAT, and MASER should also include the global state as part of the input. Is that the case in your study?

(h) The number of team skills and individual skills are crucial hyperparameters and should be explicitly stated in the main paper.

[1] Chen, Jiayu, Jingdi Chen, Tian Lan, and Vaneet Aggarwal. "Scalable multi-agent covering option discovery based on Kronecker graphs." Advances in Neural Information Processing Systems 35 (2022): 30406-30418.

**Limitations:**

The limitation part is not included.

---

> ### Author Rebuttal · Authors · 2023-08-09
>
> We thank the reviewer for the detailed comments. We hope we can address your concerns below.
>
> **Q1**: Weaknesses (a) in the Official Review.
>
> **A1**: Thank you for pointing out our potential limitations. For (1), as mentioned in lines 252-258, HMASD performs only one timestep of centralized execution in every $k$ timesteps. Such a small number of centralized information introduced can coordinate agents better from a global view compared to the fully decentralized method. And in many applications, it allows such a small amount of centralized execution. For (2), our team skills are for the entire team and expected to explore different global state spaces. it will be more difficult to learn team skills for a sub-team. How to divide the team into sub-team and how to learn team skills for sub-teams with variable number of agents. We think this could be an important direction in our future research.
>
> **Q2**: Weaknesses (b) in the Official Review.
>
> **A2**: Yes, in the early version of HMASD, we found that HMASD didn't work at all. We have iterated 28 versions to achieve the performance in the paper, including the problem formulation of multi-agent skill discovery, the way of training skills, the model structure, the hyperparameters, etc. Our future work aims to make HMASD more practical at solving sparse-reward multi-agent tasks.
>
> **Q3**: The statement in Line 40 lacks clarity. Could you expand on why the complexity can be high? Since your work also aims to discover multi-agent skills, how have you addressed this complexity issue compared to [9, 10]?
>
> **A3**: The joint state and action spaces of multi-agent tasks increase exponentially with the number of agents, so directly let all agents learn joint behavior to form team skills is exponential complexity. Our method reduces this complexity by decomposing the team skill into different individual skills (easier to learn) and guiding the joint behavior of all agents to form the team skill.
>
> **Q4**: There are additional works on multi-agent option (also known as skill) discovery, such as [1]. Please consider offering a comprehensive review of this research field to highlight your novel contributions.
>
> **A4**: Thanks for your suggestion. We will conduct a wider review of this research field in our next revision.
>
> **Q5**:  it seems that the agents in the environment must be homogeneous as they share the same individual skill space. Is that correct?
>
> **A5**: No, our method can also deal with heterogeneous agents. Although all agents share a same set of individual skills, the set of individual skills can be seen as the union of all agents' individual skills. An agent is not required to use all individual skills in the individual skill space.
>
> **Q6**: In Line 197, it's unclear how the protocol design can prevent skill duplication.
>
> **A6**: In some multi-agent tasks, it has the need for assigning different skills to agents with similar observations. In muti-agent skill learning methods like [1] and [2], each agent selects skills only based on its local observation, leading to potential skill duplication between agents with similar observations. In our method, agents select skills sequentially. When an agent selects the skill, it will know all previous agents' selected skills, which can prevent skill duplication between agents with similar observations.
>
> **Q7**: It's unclear how you can maximize the lower bound in Eq. 3 with the objectives in Eq. 4-8.
>
> **A7**: The lower bound in Eq. 3 has four terms, i.e, team reward, diversity term, skill entropy term and action entropy term. We maximize these four terms with different components in HMASD. We maximize the team reward by using it as the extrinsic reward for both the high-level policy and low-level policy. We maximize the diversity term by using it as the intrinsic reward for the low-level policy. The skill entropy term and action entropy term are maximized by optimizing the entropy of high-level policy and low-level policy, respectively. We also introduce several coefficients to balance the optimization weights of the four terms.
>
> **Q8**: For fair comparisons, MAPPO, MAT, and MASER should also include the global state as part of the input. Is that the case in your study?
>
> **A8**: We used the global state for MAPPO and MASER. Due to the problem formulation of MAT, we can't find a place to use global state in MAT. Besides, even if the environments don't provide global state, we can concentrate all agents' observations as the global state, which is the way we used in Overcooked and it can also get good performance.
>
> **Q9**: The number of team skills and individual skills are crucial hyperparameters and should be explicitly stated in the main paper.
>
> **A9**: Due to page limit, we list the number of skills on all scenarios in Table 3 of Appendix F. We will consider putting it in the main paper in our next revision.
>
> **Q10**: The limitation part is not included.
>
> **A10**: HMASD has two main limitations. Firstly, as mentioned in lines 320-321, when the state-observation space is large, HMASD can discover diverse skills but maybe only some of them (about $24$% on SMAC) are useful for the team reward. There are a large percentage of skills are assigned to explore those zero-reward states. Secondly, as mentioned in line 52 of Appendix E, the number of team skills and individual skills should be carefully adjusted. When applying HMASD to a new task, it require to adjust several hyperparameters as shown in Table 3 of Appendix F. Our future work aims to improve these limitations and make HMASD better at solving sparse-reward multi-agent tasks.
>
> Finally, thank you again for your recognition and insightful review to our work. We will incorporate your suggestions into our next revision.
>
> ##### Reference
>
> [1] J. Yang, I, et al. Hierarchical cooperative multi-agent reinforcement learning with skill discovery.
>
> [2] Y. Liu, et al. Heterogeneous skill learning for multi-agent tasks.

---

> > ### Comment · Reviewer_Ty5M · 2023-08-19
> >
> > Thanks for the detailed rebuttal. This has resolved most of my concerns. I look forward to seeing improvements related to Weakness (a) in the future version of this paper. I will maintain my current score.

---

### Official Review · Reviewer_HRuC · 2023-06-28

**Soundness:** 3 good
**Presentation:** 3 good
**Contribution:** 3 good
**Rating:** 5
**Confidence:** 5

**Summary:**

The paper proposes a two level hierarchical model for cooperative multi-agent RL. The key idea is to use variational inference based skill discovery over joint and individual policies. Intuitively, the objective can described as follows: i) find individual options, that are diverse (in terms of state visitations), ii) find joint options that are diverse (in terms of joint state visitation) iii) maximise the reward.
The experimental results on several cooperative domains are presented and the method performs better than well established baselines.

**Strengths:**

- The paper presents a technically solid, novel algorithm.
- The experiments are convincing and demonstrate that the method indeed discovers helpful joint and individual skills and combines them to into a reward-maximising policy


**Weaknesses:**

- My main concern is that most of the improvement comes from implicit exploration bonus that arises from skill discovery objective, rather than from decomposition of the main task into subs tasks. It would be more convincing to have some sort of exploration bonus baseline. For example, adding a reward for individuals visiting new states and population visiting new joint states. For example, one could derive them via "Exploration by random network distillation" (https://arxiv.org/abs/1810.12894) method, one RND trained on individual states and another on the joint and summing up the reward.

- There is no discussion on limitations.

- Minor issue. The notation is a bit overloaded and makes things slightly confusing:
 - line 143-145. $Z \in \mathcal{Z}$ and $z \in \mathcal{X}$ is confusing
 - line 157-159. p and q seem to be referring to two distributions each




**Questions:**

I would like the authors to address the comment above on the (joint) exploration baseline.
I would also like the authors to explicitly discuss the limitations of the method.

I am willing to increase my score based on the answers.

**Limitations:**

There is no discussion on limitations, which I would encourage the authors to write.

---

> ### Author Rebuttal · Authors · 2023-08-09
>
> We thank the reviewer for the detailed comments. We hope we can address your concerns below.
>
> **Q1**: My main concern is that most of the improvement comes from implicit exploration bonus that arises from skill discovery objective, rather than from decomposition of the main task into subs tasks. It would be more convincing to have some sort of exploration bonus baseline.
>
> **A1**: Thanks for your suggestion. Here, we compare HMASD with an exploration bonus MARL baseline, EITI/EDTI[1], on Overcooked. EITI/EDTI proposes exploration strategies where agents start with decentralized exploration driven by their individual curiosity (i.e., individual exploration), and are also encouraged to coordinate their exploration (i.e., joint exploration). The final reward for each agent is the sum of team reward, individual exploration bonus and joint exploration bonus. EITI and EDTI provide 2 ways to calculate the joint exploration bonus. The comparison of HMASD with EITI/EDTI on Overcooked is shown in Table 1.
>
> Table1: The final performance of episode reward on Overcooked.
>
> |       Scenario        |       EITI        |       EDTI        |      HMASD       |
> | :------------------- | :--------------- | :--------------- | :-------------- |
> |     cramped_room      | 0.122 $\pm$ 0.013 | 0.119 $\pm$ 0.015 | 236.0 $\pm$ 8.0  |
> | asymmetric_advantages | 0.008 $\pm$ 0.002 | 0.004 $\pm$ 0.002 | 295.1 $\pm$ 84.5 |
> |   coordination_ring   |   0.0 $\pm$ 0.0   |   0.0 $\pm$ 0.0   | 80.0 $\pm$ 97.9  |
>
> We can see that HMASD outperforms EITI and EDTI on all Overcooked scenarios. Although EITI/EDTI encourages both individual exploration and joint exploration, it doesn't formulate the relationship between individual exploration and joint exploration. These two explorations may have conflicts, causing neither exploration to work well. In our work, we propose to discover both team and individual skills for solving sparse-reward multi-agent tasks. Importantly, we build a probabilistic graphical model to formulate the relationship between team skill and individual skill, and then derive a lower bound as our objective. The results show that our method could discover significant team and individual skills, and effectively combine them to accomplish the sparse-reward multi-agent task.
>
> **Q2**: There is no discussion on limitations.
>
> **A2**: HMASD has two main limitations. Firstly, as mentioned in lines 320-321, when the state-observation space is large, HMASD can discover diverse skills but maybe only some of them (about $24$% on SMAC) are useful for the team reward. There are a large percentage of skills are assigned to explore those zero-reward states. Secondly, as mentioned in line 52 of Appendix E, the number of team skills and individual skills should be carefully adjusted. When applying HMASD to a new task, it require to adjust several hyperparameters as shown in Table 3 of Appendix F. Our future work aims to improve these limitations and make HMASD better at solving sparse-reward multi-agent tasks.
>
> **Q3**: Minor issue. The notation is a bit overloaded and makes things slightly confusing: line 143-145. $Z \in \mathcal{Z}$ and $z^i \in \mathcal{X}$ is confusing; line 157-159. p and q seem to be referring to two distributions each
>
> **A3**: (1) $\mathcal{Z}$ is the team skill space and $\mathcal{X}$ is the individual skill space. The team skill space is for the whole team, and the individual skill space is for each individual agent. All agents share a same individual skill space $\mathcal{X}$. (2) In this paper, we adopt the structured variational inference to derive the lower bound. In structured variational inference, we aim to approximate some distribution $p(y)$ with another, potentially simpler distribution $q(y)$. Typically, $q(y)$ is taken to be some tractable factorized distribution. So $p$ is the true distribution and $q$ is the approximate distribution for $p$.
>
> Finally, thank you again for your thoughtful comments. We will incorporate your suggestions into our next revision. If some of your concerns are addressed, you could consider raising the rating. This is very important for us and we will appreciate it very much.
>
> ##### Reference
>
> [1] Wang, T, et al. Influence-Based Multi-Agent Exploration. ICLR 2020.

---

### Official Review · Reviewer_7A65 · 2023-07-01

**Soundness:** 3 good
**Presentation:** 3 good
**Contribution:** 3 good
**Rating:** 6
**Confidence:** 3

**Summary:**

This paper proposed HMASD, a two-level hierarchical algorithm for discovering both team and individual skills in MARL. The high-level policy based on the transformer structure generates team skills and individual skills in an autoregressive manner, and the low-level policies output primitive actions according to individual skills and local observations. The authors formulate multi-agent skill discovery as an inference problem by augmenting the basic probabilistic graphical model. Experimental results show that HMASD can outperform other baselines in sparse reward multi-agent tasks.

**Strengths:**

1. The paper is well-organized and well-motivated. The authors explain their formalism extremely well throughout, including in their methods section.
2. The authors design a toy game *Alice_and_Bob* to demonstrate how their method works, which improves the soundness of their method.
3. Some MARL work related to skill discovery or exploration is fully mentioned in the appendix.
4. The authors conduct solid experiments in some popular benchmarks, and carry out sufficient ablation experiments.

**Weaknesses:**

1. Some important baselines are missing in the experiment section, such as HSD and CMAE.
2. Some curves are stopped while learning does not seem to have converged in Figure 6 & 7.
3. Due to the introduction of more hyperparameters (8 new hyperparameters can be seen from Table 3), HMASD needs more hyperparameter tuning.

**Questions:**

1. Shouldn't the distribution over $\mathcal{O}$ be proportional to $\exp(r(s,\boldsymbol{a}))$ in L121?
2. Would the initialization method of the arbitrary symbol $Z_0$ affect the performance of HMASD?
3. Is the build order of the individual skills preset?
4. In the *Alice_and_Bob* game, does $Z=1$ always correspond to the team skill collecting the red diamond?
5. Can HMASD be applied to value-based multi-agent reinforcement learning algorithms such as QMIX?
6. How does HMASD perform compared to CMAE and EITI/EDTI [1]?

**Reference**

[1]  Wang, Tonghan et al. Influence-Based Multi-Agent Exploration. 2019.

**Limitations:**

The authors state that the limit of their method is the number of team skills and individual skills should be carefully adjusted. That is to say, due to the introduction of more hyperparameters, HMASD needs more hyperparameter tuning.

---

> ### Author Rebuttal · Authors · 2023-08-09
>
> We thank the reviewer for the detailed comments. We hope we can address your concerns below.
>
> **Q1**: Some important baselines are missing in the experiment section, such as HSD and CMAE.
>
> **A1**: HSD is an old method proposed in 2019. It performs poorly even on the dense reward SMAC as shown in [1]. In this paper, we select most recent related works as our baselines, including MAT(NeurIPS 2022)，MAPPO(NeurIPS 2022)，MASER(ICML 2022). Besides, HSD is a hierarchical MARL method that only learns individual skills, which has similar idea with HMASD\_NoTeam in our ablations. We have compared HMASD with HMASD\_NoTeam to verify the importance of learning team skills. So we don't use HSD as our baseline. CMAE is a solid method but has complex implementation details. We don't choose CMAE as a baseline because CMAE doesn't provide the code running on SMAC. There are someone that raised an issue about the SMAC running code on its github repository, but the author did not reply.
>
> **Q2**: Some curves are stopped while learning does not seem to have converged in Figure 6 & 7.
>
> **A2**: Yes, we stop the training when HMASD converges. Overall, HMASD can achieve faster convergence and higher average performance than baselines.
>
> **Q3**: Due to the introduction of more hyperparameters (8 new hyperparameters can be seen from Table 3), HMASD needs more hyperparameter tuning.
>
> **A3**: Yes, this is one limitation of HMASD. We mainly adjust hyperparameters through grid-search. We don't directly perform grid-search on all 8 hyperparameters. We divide the 8 hyperparameters into 3 groups, i.e., $(k, n_Z, n_z), (\lambda_h, \lambda_l), (\lambda_e, \lambda_D, \lambda_d)$. We first perform grid-search on $(k, n_Z, n_z)$ and fix them. The next grid-search is on $(\lambda_e, \lambda_D, \lambda_d)$ and the last is on $(\lambda_h, \lambda_l)$. Although the final hyperparameters obtained in this way may be not the best,  it can greatly reduce tuning time.
>
> **Q4**: Shouldn't the distribution over $\mathcal{O}$ be proportional to exp⁡($r(s,\boldsymbol{a})$) in L121?
>
> **A4**: We follow the Eq. (3) in [2] to define the distribution over $\mathcal{O}$. Even if the distribution over $\mathcal{O}$ is proportional to exp⁡($r(s,\boldsymbol{a})$), it just adds a constant to the optimized lower bound in Eq. (3), which has no effect on our method.
>
> **Q5**: Would the initialization method of the arbitrary symbol $Z_0$ affect the performance of HMASD?
>
> **A5**: Following MAT, we set the first symbol of the decoder (i.e., $Z_0$) to a fixed vector. We would like to explore the effect of $Z_0$ in the future.
>
> **Q6**: Is the build order of the individual skills preset?
>
> **A6**: No, all individual skills are learned equivalently, not learned in a specific order. The learning of individual skills for each agent depends on the skill assignment of the high-level skill coordinator. And we train skill coordinator, discover, discriminator simultaneously.
>
> **Q7**: In the *Alice_and_Bob* game, does $Z=1$ always correspond to the team skill collecting the red diamond?
>
> **A7**: No, HMASD can discover diverse team skills in different runs on *Alice_and_Bob*. We choose one of the runs that have well-explainable behaviors to visualize the skills.
>
> **Q8**: Can HMASD be applied to value-based multi-agent reinforcement learning algorithms such as QMIX?
>
> **A8**: No, it can't. From the derived lower bound in Eq. (3), we can see that the skill entropy term and the action entropy term are related to the entropy of high-level policy and low-level policy. So HMASD can only be applied to policy-based MARL.
>
> **Q9**: How does HMASD perform compared to CMAE and EITI/EDTI?
>
> **A9**: No comparison with CMAE has been explained in A1.  Here, we compare HMASD with EITI/EDTI on Overcooked. EITI/EDTI proposes exploration strategies where agents start with decentralized exploration driven by their individual curiosity (i.e., individual exploration), and are also encouraged to coordinate their exploration (i.e., joint exploration). The final reward for each agent is the sum of team reward, individual exploration bonus and joint exploration bonus. EITI and EDTI provide 2 ways to calculate the joint exploration bonus. The comparison of HMASD with EITI/EDTI on Overcooked is shown in Table 1.
>
> Table1: The final performance of episode reward on Overcooked.
>
> |Scenario|       EITI        |       EDTI        |      HMASD       |
> | :------------------- | :--------------- | :--------------- | :-------------- |
> |     cramped_room      | 0.122 $\pm$ 0.013 | 0.119 $\pm$ 0.015 | 236.0 $\pm$ 8.0  |
> | asymmetric_advantages | 0.008 $\pm$ 0.002 | 0.004 $\pm$ 0.002 | 295.1 $\pm$ 84.5 |
> |   coordination_ring   |   0.0 $\pm$ 0.0   |   0.0 $\pm$ 0.0   | 80.0 $\pm$ 97.9  |
>
> We can see that HMASD outperforms EITI and EDTI on all Overcooked scenarios. Although EITI/EDTI encourages both individual exploration and joint exploration, it doesn't formulate the relationship between individual exploration and joint exploration. These two explorations may have conflicts, causing neither exploration to work well. In our work, we propose to discover both team and individual skills for solving sparse-reward multi-agent tasks. Importantly, we build a probabilistic graphical model to formulate the relationship between team skill and individual skill, and then derive a lower bound as our objective. The results show that our method could discover significant team and individual skills, and effectively combine them to accomplish the sparse-reward multi-agent task.
>
> Finally, thank you again for your thoughtful comments. We will incorporate your suggestions into our next revision. If some of your concerns are addressed, you could consider raising the rating. This is very important for us and we will appreciate it very much.
>
> Reference
>
> [1] Wang T, et al. Rode: Learning roles to decompose multi-agent tasks.
>
> [2] Levine S. Reinforcement learning and control as probabilistic inference: Tutorial and review.

---

> > ### Comment · Reviewer_7A65 · 2023-08-13
> > **Thank you for covering my questions.**
> >
> > I appreciate the effort the authors have put into addressing my concerns. Their responses have effectively alleviated the concerns I had initially. As a result, I have chosen to revise my ratings for this paper. Thank you for the thorough and satisfactory replies.

---

### Official Review · Reviewer_skAV · 2023-07-06

**Soundness:** 3 good
**Presentation:** 4 excellent
**Contribution:** 3 good
**Rating:** 6
**Confidence:** 3

**Summary:**

This paper focuses on applying unsupervised skill learning to multi-agent reinforcement learning. For this purpose, the authors proposed a two-level hierarchical algorithm for discovering both team and individual skills in MARL, where individual skills refers to the abilities of individual agents and team skills refer to the ability of agents to work together as a whole. To this end, they embed the multi-agent skill discovery problem into a probabilistic graphical model and formulate it as an inference problem. Finally, they show that the proposed method achieves superior performance on sparse reward MARL benchmarks.

**Strengths:**

1. The problem this paper considers is rather important and it is a promising way to learn a set of skills and combine them properly to tackle complex tasks.
2. The literature review is sufficient in Appendix B.
3. The proposed method of decomposing the team skill into different individual skills for agents and ensuring that the joint behavior of all agents can form the team tactic is well-motivated with the football example.
4. The proposed method is novel as this work is the first attempt to model both team skills and individual skills with the probabilistic graphical model in MARL.
5. The empirical evaluation, especially Figure 4, is of high quality and quite interesting.
6. The results on SMAC with sparse rewards and Overcooked are significant.
7. The paper is generally well-written.

**Weaknesses:**

The reviewer is concerned about the training of the proposed method. (1) Too many components that require function approximation may bring instability into the MARL training process.  (2) In Figure 3 of Appendix E, the proposed method seems to be very sensitive to hyperparameters.

**Questions:**

1. Have the authors encountered instability problems when training as there are so many components combined with the MARL algorithm?
2. In Table 3 of Appendix F, different tasks require different hyperparameters. How did the authors choose these hyperparameters? Is it possible to find a suitable set of hyperparameters that can be suitable for most tasks?

**Limitations:**

This paper does not discuss the limitations. The biggest limitation might be that this method is too sensitive to hyperparameters, making it difficult to apply it directly to new tasks.

---

> ### Author Rebuttal · Authors · 2023-08-09
>
> We thank the reviewer for the detailed comments. We hope we can address your concerns below.
>
> **Q1**: Have the authors encountered instability problems when training as there are so many components combined with the MARL algorithm?
>
> **A1**: Yes, in the early version of HMASD, we found that HMASD performed poorly even in the dense reward tasks. We have iterated 28 versions to achieve the performance in the paper, including the problem formulation of multi-agent skill discovery, the way of training skills, the model structure, the hyperparameters, etc.
>
> **Q2**: In Table 3 of Appendix F, different tasks require different hyperparameters. How did the authors choose these hyperparameters? Is it possible to find a suitable set of hyperparameters that can be suitable for most tasks?
>
> **A2**: We mainly adjust hyperparameters through grid-search. There are 8 important hyperparameters in HMASD as shown in Table 3 of Appendix F. We don't directly perform grid-search on all 8 hyperparameters. We divide the 8 hyperparameters into 3 groups, i.e., $(k, n_Z, n_z), (\lambda_h, \lambda_l), (\lambda_e, \lambda_D, \lambda_d)$. We first perform grid-search on $(k, n_Z, n_z)$ and fix them. The next grid-search is on $(\lambda_e, \lambda_D, \lambda_d)$ and the last is on $(\lambda_h, \lambda_l)$. Although the final hyperparameters obtained in this way may be not the best,  it can greatly reduce tuning time. Among 8 hyperparameters, we find that the skill interval $k$, the number of team skills $n_Z$ and the number of individual skills $n_z$ are the three most important hyperparameters. The ablations on the three hyperparameters can be found in Appendix E. The performance of HMASD varies greatly with different $k, n_Z, n_z$. These hyperparameters mainly depend on the specific task, especially the number of skills for solving the task. It is possible to find a suitable set of hyperparameters that can generalize to a set of similar tasks.
>
> **Q3**: This paper does not discuss the limitations. The biggest limitation might be that this method is too sensitive to hyperparameters, making it difficult to apply it directly to new tasks.
>
> **A3**: HMASD has two main limitations. Firstly, as mentioned in lines 320-321, when the state-observation space is large, HMASD can discover diverse skills but maybe only some of them (about $24$% on SMAC) are useful for the team reward. There are a large percentage of skills are assigned to explore those zero-reward states. Secondly, as mentioned in line 52 of Appendix E, the number of team skills and individual skills should be carefully adjusted. When applying HMASD to a new task, it require to adjust several hyperparameters as shown in Table 3 of Appendix F. Our future work aims to improve these limitations and make HMASD better at solving sparse-reward multi-agent tasks.
>
> Finally, thank you again for your recognition and positive review to our work. We will incorporate your suggestions into our next revision.

---

### Official Review · Reviewer_Jm7Q · 2023-07-11

**Soundness:** 2 fair
**Presentation:** 3 good
**Contribution:** 2 fair
**Rating:** 5
**Confidence:** 2

**Summary:**

This paper presents Hierarchical Multi-Agent Skill Discovery (HMASD) that can discover both team and individual skills in MARL. The authors formulate multi-agent skill discovery as an inference problem in probabilistic graphical models. The model consists of a skill coordinator that reasons about team and individual skills, a skill discoverer that maps skills into actual execution, and a skill discriminator that encourages the learning of diverse and distinguishable skills. The proposed method is evaluated on sparse reward multiagent benchmark including SMAC and overcooked and is shown to achieve superior performance comparing to baselines. Ablation studies are also done to verify the effectiveness of each proposed components.

**Strengths:**

- The proposed method is intersting.
- The paper is nicely structured.

**Weaknesses:**

There is a bit limited given only training curves are shown. More fine-grained experiments like visualization of learned skills, progression of skill learning, etc., may provide more insight into the effectiveness of the proposed method. Overall the experiment is on a low side and see more detailed comments in the questions section.

**Questions:**

- In figure 6, there is no orange line and does that mean MAPPO fails to learn anything?
- By comparing figure 6 and 7, the performance of MASER is so different. In figure 6, MASER is the only baseline that works; however, in figure 7, MASER is the baseline that works the worst. Could authors briefly explain why this is the case?
- Why conduct experiment in SMAC rather than SMACv2?
- I cannot find the 3m scenario in SMAC [40]. Does it mean 3 marine or MMM2?
- How do the authors deal with the heterogenous agents in SMAC 2s_vs_1sc and 2m_vs_1z?
- It would be interesting to visualize the learned skills in both SMAC and overcooked environments.
- In the paragraph at line 314, the authors discuss interesting observation of how skill learning is done. It will be more convincing if some quantitative results can be shown here to support the statement.
- In spite of the limited space, it's better to include the related work section in the main paper to make it more self-contained.
- The variances of HMASD in 2s_vs_1sc, 2m_vs_1z, and coordination_ring are extremely large and the lower bounds of the shaded area are at zero, making it hard to justify the effectiveness of the method. What is the percentage of the model to learn meaningful behavior among all trials?
- How many random seeds are used for plotting those training curves?

**Limitations:**

The limitation of this work is not explicitly discussed in the paper.

---

> ### Author Rebuttal · Authors · 2023-08-09
>
> We thank the reviewer for the detailed comments. We hope we can address your concerns below.
>
> **Q1**: In figure 6, there is no orange line and does that mean MAPPO fails to learn anything?
>
> **A1**: Yes, the orange line is covered by the blue and green lines. Both MAT and MAPPO fail to learn anything on SMAC with 0-1 reward.
>
> **Q2**: In figure 6, MASER is the only baseline that works; however, in figure 7, MASER is the baseline that works the worst. Could authors briefly explain why this is the case?
>
> **A2**: MASER proposes to automatically generate subgoals for agents, and it only demonstrates the effectiveness on the SMAC tasks in the original paper. In our work, we find that MASER works only in SMAC but fails in Overcooked. There are other works[1,2] that also find that MASER performs poorly on other multi-agent tasks. This may be because the automatic subgoal generation mechanism of MASER may only work in some specific tasks, e.g., SMAC.
>
> **Q3**: Why conduct experiment in SMAC rather than SMACv2?
>
> **A3**: SMACv2 was proposed because SMAC with dense reward is so easy that many algorithms (such as MAT and MAPPO) can achieve almost 100% win rate on all dense-reward SMAC scenarios. In our work, we aim to solve challenging multi-agent tasks with sparse reward. And in our early experiments,  we found that SMAC with 0-1 reward is hard enough to compare the performance of different algorithms. So we choose SMAC rather than SMACv2.
>
> **Q4**: I cannot find the 3m scenario in SMAC. Does it mean 3 marine or MMM2?
>
> **A4**: The original SMAC paper introduces only 14 scenarios, while SMAC's codebase provides more scenarios, including 3m. The 3m scenario represents 3 allied marines versus 3 enemy marines.
>
> **Q5**: How do the authors deal with the heterogeneous agents in SMAC 2s_vs_1sc and 2m_vs_1z?
>
> **A5**: In both 2s_vs_1sc and 2m_vs_1z, the type of all agents controlled by MARL method is the same. Therefore, they are scenarios with homogeneous agents, not heterogeneous agents. Our method can also deal with heterogeneous agents.
>
> **Q6**: It would be interesting to visualize the learned skills in both SMAC and overcooked environments.
>
> **A6**: Thanks for your suggestion. Due to time limit, we visualize the learned individual skills on the SMAC scenario 3m as shown in Figure 1, which can be found in the one-page PDF of the global response. We can observe that only 1 of 3 skills can result in a non-zero team reward, and the remaining 2 skills explore the state-observation spaces that don’t contribute to the team reward.  We will provide more skill visualizations in our next revision.
>
> **Q7**: In the paragraph at line 314, the authors discuss interesting observation of how skill learning is done. It will be more convincing if some quantitative results can be shown here to support the statement.
>
> **A7**: We count the average percentage of useful individual skills among all learned individual skills on the SMAC. For each SMAC scenario, we conduct $5$ different runs with different random seeds. We learn $3, 5, 2$ individual skills for each run on 3m, 2s_vs_1sc, 2m_vs_1z, respectively. Therefore, we learn $3\*5+5\*5+2\*5=50$ individual skills on three SMAC scenarios. After our test, only $12$ individual skills are useful for the team performance. In other words, only $24$% of the learned individual skills are useful for completing the task on average after training.
>
> **Q8**: In spite of the limited space, it's better to include the related work section in the main paper to make it more self-contained.
>
> **A8**: Thanks for your suggestion. We will include the related work section in the main paper in our next revision.
>
> **Q9**: The variances of HMASD in 2s_vs_1sc, 2m_vs_1z, and coordination_ring are extremely large and the lower bounds of the shaded area are at zero, making it hard to justify the effectiveness of the method. What is the percentage of the model to learn meaningful behavior among all trials?
>
> **A9**: For every scenario, we conduct $5$ runs. For each run on the SMAC scenario, once agents discover useful skills, the performance will quickly increase to $1$. So the final performance is either $1$ or $0$, which leads to large variance among $5$ runs. There are $7$ scenarios in our paper. Among $5*7=35$ runs, $26$ runs could learn meaningful behavior. Therefore, the percentage of the model to learn meaningful behavior among all trials is $26/35=74.3$%. One of our future goals is to increase this percentage and reduce the variances of our method.
>
> **Q10**: How many random seeds are used for plotting those training curves?
>
> **A10**: As mentioned in line 268, for each training curve, we show the mean and variance of the performance across five different random seeds.
>
> **Q11**: The limitation of this work is not explicitly discussed in the paper.
>
> **A11**: HMASD has two main limitations. Firstly, as mentioned in lines 320-321, when the state-observation space is large, HMASD can discover diverse skills but maybe only some of them (about $24$% on SMAC) are useful for the team reward. There are a large percentage of skills are assigned to explore those zero-reward states. Secondly, as mentioned in line 52 of Appendix E, the number of team skills and individual skills should be carefully adjusted. When applying HMASD to a new task, it require to adjust several hyperparameters as shown in Table 3 of Appendix F. Our future work aims to improve these limitations and make HMASD better at solving sparse-reward multi-agent tasks.
>
> Finally, thank you again for your thoughtful comments. We will incorporate your suggestions into our next revision. If some of your concerns are addressed, you could consider raising the rating. This is very important for us and we will appreciate it very much.
>
> ##### Reference
>
> [1] Yang X, et al. Learning Graph-Enhanced Commander-Executor for Multi-Agent Navigation.
>
> [2] Li W, et al. Semantically Aligned Task Decomposition in Multi-Agent Reinforcement Learning.

---

> > ### Comment · Reviewer_Jm7Q · 2023-08-17
> > **Thanks for the rebuttal**
> >
> > I appreciate the authors' effort on the rebuttal.
> >
> > Overall, the rebuttal addressed most of my concern. Still, I believe the paper will be much stronger with more fine-grained analysis like A6, A7, A8 along with results requested by the other reviewers. Due to the time limit of the rebuttal, I am satisfied with the additional experiments; but adding more thorough and extensive results for all new experimental analysis should be done in the revision. Hence, I will increase my rating.

---

> > > ### Author Response · Authors · 2023-08-18
> > > **Thanks for your reply**
> > >
> > > We are happy that we could address most of your concerns. We promise to incorporate our rebuttal results and add more fine-grained analysis in the next revision.

---

### Author Rebuttal · Authors · 2023-08-09

We have uploaded a one-page PDF containing a new figure that visualizes the learned individual skills on the SMAC scenario 3m.

---

### Decision · Program_Chairs · 2023-09-21

**Decision:**

Accept (poster)

**Comment:**

This paper proposes a hierarchical model for cooperative multi-agent reinforcement learning. The paper presents a novel algorithm that discovers both team and individual skills for solving sparse-reward multi-agent tasks. The reviews generally praise the technical contributions and experimental results, but also point out concerns and weaknesses, such as the complexity of the framework and the need for more clarity and details in certain areas. The authors provide rebuttals that address these concerns. Overall, the reviews suggest the paper to be of moderate-to-high impact, meanwhile there are still minor revisions the author should take care of.